# Uncertainty Assessment of Entropy-Based Circular Channel Shear Stress Prediction Models Using a Novel Method

**Amin Kazemian-Kale-Kale [1], Azadeh Gholami [1], Mohammad Rezaie-Balf [2], Amir Mosavi [3,4,5], Ahmed A. Sattar [6], Amir H. Azimi [7], Bahram Gharabaghi [8] and Hossein Bonakdari [9,\***

[1] Environmental Research Centre, Razi University, Kermanshah 6718773654, Iran; aminkazemi_akkk@yahoo.com (A.K.-K.-K.); gholamiazadeh1@gmail.com (A.G.)
[2] Department of Water Engineering, Graduate University of Advanced Technology, Kerman 7631818356, Iran; moe.rezaie69@gmail.com
[3] Department of Mathematics, J. Selye University, 94501 Komarno, Slovakia; amir.mosavi@kvk.uni-obuda.hu
[4] Department of Automation, Obuda University, 1034 Budapest, Hungary
[5] Faculty of Civil Engineering, Technische Universität Dresden, 01069 Dresden, Germany
[6] Department of Irrigation & Hydraulics, Faculty of Engineering, Cairo University, Cairo 12613, Egypt; ahmoudy77@yahoo.com
[7] Department of Civil Engineering, Lakehead University, Thunder Bay, ON P7B 5E1, Canada; azimi@lakeheadu.ca
[8] School of Engineering, University of Guelph, Guelph, ON NIG 2W1, Canada; bgharaba@uoguelph.ca
[9] Department of Soils and Agri-Food Engineering, Laval University, Quebec City, QC G1V 0A6, Canada
\* Correspondence: hossein.bonakdari@fsaa.ulaval.ca; Tel.: +1-418-656-2131; Fax: +1-418-656-3723

**Abstract:** Entropy models have been recently adopted in many studies to evaluate the shear stress distribution in open-channel flows. Although the uncertainty of Shannon and Tsallis entropy models were analyzed separately in previous studies, the uncertainty of other entropy models and comparisons of their reliability remain an open question. In this study, a new method is presented to evaluate the uncertainty of four entropy models, Shannon, Shannon-Power Law (PL), Tsallis, and Renyi, in shear stress prediction of the circular channels. In the previous method, the model with the largest value of the percentage of observed data within the confidence bound ($N_{in}$) and the smallest value of Forecasting Range of Error Estimation (FREE) is the most reliable. Based on the new method, using the effect of Optimized Forecasting Range of Error Estimation (FREE$_{opt}$) and Optimized Confidence Bound (OCB), a new statistic index called FREE$_{opt}$-based OCB (FOCB) is introduced. The lower the value of FOCB, the more certain the model. Shannon and Shannon PL entropies had close values of the FOCB equal to 8.781 and 9.808, respectively, and had the highest certainty, followed by $\rho g R s$ and Tsallis models with close values of 14.491 and 14.895, respectively. However, Renyi entropy, with the value of FOCB equal to 57.726, had less certainty.

**Keywords:** water resources; uncertainty; shear stress distribution; circular channel; entropy; Shannon; Shannon-Power Law (PL); Tsallis; Renyi

## 1. Introduction

In circular open channels, shear stress distribution has been experimentally studied by many researchers [1–3]. Numerical and analytical models have also been developed to predict shear stress distribution along channels [4–7]. Recently, soft computing enabled researchers to estimate shear stress distribution in open channels [8–16].

The novel application of the entropy theory has been successful in modelling velocity distribution [17–19], the transverse slope of stable channels banks [20,21], and shear stress distribution [22]. For the first time, Chiu introduced models for estimating the shear stress and velocity distribution in open channels using the Shannon entropy concept [23]. Sterling and Knight developed equations based on Shannon entropy for predicting boundary shear stress [24]. However, their study showed limitations in reflecting some hydraulic

characteristics of flow in open channels, but their results indicated that this method could reasonably well predict the shear stress distribution. Sheikh and Bonakdari employed the Shannon entropy concept and Power Law (PL) techniques to develop new equations for predicting shear stress distribution [25]. Their results compared with measured data and showed that their proposed Shannon PL model had good potential for practical applications besides Shannon entropy.

Renyi entropy is defined as a generalized form of entropy that was introduced by Renyi [26]. Renyi entropy can be considered a generalization of the Shannon entropy, as Shannon entropy is a particular case of Renyi entropy [27,28]. Tsallis proposed an entropy as a generalization of the Shannon entropy comprising a supplementary parameter called "Tsallis entropy" [29]. Tsallis entropy with non-additive parameters are less susceptible to the form of the probability distribution [30]. Tsallis and Renyi entropies were applied to characterize threshold channel bank profiles and sediment concentration distribution [31,32].

Bonakdari et al. showed that the Tsallis method is suitable for the calculation of shear stress distribution along the wetted perimeter with reasonable accuracy [33]. Khozani and Bonakdari employed the Renyi entropy and presented a novel prediction model [22]. Although many studies were carried out on entropy models, wide implementation of entropy models has not taken place due to the absence of enough confidence in these models compared to previous conventional models in estimating flow variables. Thus, it is beneficial to analyze the uncertainty of entropy models and compare their performance with previous traditional models.

The uncertainty analysis of many hydraulic and hydrological models has been tackled by many researchers [34–47]. Thiemann et al. developed a Bayesian formulation, which permits the hydrologist to quantify uncertainty [48]. The method is called Bayesian Recursive Estimation (BaRE).

Misirli et al., using BaRE and Monte-Carlo simulation (BMC), presented an uncertainty method to analyze streamflow uncertainty [36]. The uncertainty method of Misirli et al. was applied to the uncertainty analysis of the Shannon entropy model in estimating the velocity distribution [49]. Kazemian-Kale-Kale et al., with the improved uncertainty method used in previous studies [32,36,44,45,48,49], analyzed uncertainty of the Tsallis entropy model in predicting shear stress distribution [50]. They emphasized the necessity to normalize the shear stress data for the uncertainty analyses. Then, they calibrated the model to select the best sample size (shear stress data considered under different hydraulic conditions) and finally analyzed the uncertainty of the Tsallis entropy model using this sample size (SS). Although their results were well capable of analyzing the uncertainty of the Tsallis model, their calibration method was difficult, and they did not discuss the performance of the different transfer functions to normalization in the uncertainty results.

Therefore, in another study, Kazemian-Kale-Kale et al. simplified the calibration method of their previous study. In addition to the Box–Cox function, Johnson's function was used to analyze the Shannon entropy uncertainty [51].

In the present study, the uncertainty of the four different entropy models of Shannon, Shannon PL, Tsallis, and Renyi are compared to predict shear stress in the circular channels. The uncertainty prediction is implemented by modifying the uncertainty method presented in previous research [51], and a novel uncertainty method is introduced. In addition, as a criterion for comparison, the uncertainty of the $\rho g R s$ model (the common model in prediction of shear stress values) is evaluated, as well. At first, the uncertainty method introduced by Kazemian-Kale-Kale et al. [51] introduced the uncertainty method based on the BMC method briefly introduced as the Hybrid Bayesian and Monte-Carlo Estimation System (HBMES-1).

In this study, the HBMES-1 method to determine whether or not with a 95% confidence bound (95%CB), entropy models are sufficiently certain to predict shear stress in circular channels. The answer to this question is determined by the percentage of measured data within the confidence bound ($N_{in}$), but the values of FREE statistics should also be checked for the accuracy of each model relative to the $N_{in}$. FREE is a statistical index equal to the

absolute sum of the measured data within the confidence bound ($|F_P|$) and the absolute sum of the measured data outside the confidence bound ($|F_N|$).

It is difficult to compare the certainty of five shear stress prediction models. I Therefore, the HBMES-1 method is further improved in this study, and the uncertainty method is presented as HBMES-2. The Minimum CB that covers all measured data, OCB (Optimized Confidence Bound), is defined by employing the HBMES-2 method. Then, based on the OCB, the FREE statistic is optimized and is called $FREE_{opt}$.

Given the value of OCB and $FREE_{opt}$, the FOCB statistic is introduced that can show the effect of all uncertainty statistics. The drawn OCB represents the width of the confidence bound, which is a quantitative statistic for estimating uncertainty. In addition, $FREE_{opt}$, which represents the absolute sum of the measured data within the OCB, is a qualitative statistic for estimating uncertainty. FOCB shows the degree of qualitative and quantitative uncertainty of shear stress models. Therefore, it is easy to make a comparison using only one statistic (FOCB) in HBMES-2 method.

## 2. Materials and Methods

### 2.1. Entropy Models

#### 2.1.1. Shannon Model

Sterling and Knight employed Shannon entropy to present a model to predict shear stress distribution in circular channels as follows [24]:

$$\tau = \frac{1}{\lambda_0} \ln\left[1 + (e^{\lambda_0 \tau_{max}} - 1)\frac{y}{P/2}\right] \quad (1)$$

where $\tau$ is the local shear stress distribution, $\tau_{max}$ is the maximum shear stress, $P$ is the wetted perimeter of the circular channel, and $y$ is the specific point in wetted perimeter, on which we want to obtain the shear stress on; value changes between 0 to $P/2$. Figure 1 shows the circular cross-section with related notations used in models.

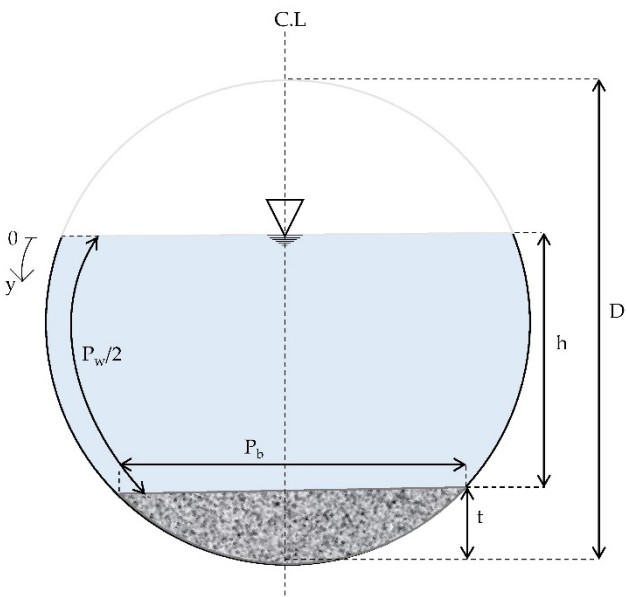

**Figure 1.** Cross-section of a circular channel section with a flat bed and its notation.

In Figure 1, $D$ is the diameter of the channel, $t$ is the height of the flat bed, and $h$ is the water depth. $P_w$ is the wetted perimeter of the channel wall and $P_b$ is the wetted perimeter of the channel bed, and their summation is equal to $P$. Therefore, the wetted perimeter of bed and wall of the channel is calculated at different heights of ($h + t/D$) ratios. The shear stress values are only computed for half the wetted perimeter due to section symmetry

and hydraulic characteristics. In Equation (1), $\lambda_0$ is the Lagrange multiplier that can be determined as [24]:

$$\frac{1}{\lambda_0} = \left[ \frac{\tau_{\max} e^{\lambda_0 \tau_{\max}}}{e^{\lambda_0 \tau_{\max}} - 1} - \rho g R s \right]^{-1} \tag{2}$$

where $\rho$ is the fluid density, $g$ is the gravity coefficient, $R$ is the hydraulic radius, and $s$ is the channel slope. In a circular channel with a flat bed, Equation (1) can be written as [24,33]:

$$\tau_w = \frac{1}{\lambda_{0w}} \left[ 1 + \left( e^{\lambda_{0w} \tau_{\max(w)}} - 1 \right) \frac{2(y - y_w)}{P_w} \right] \qquad y_w \leq y \leq \frac{P_w}{2} \tag{3}$$

$$\tau_b = \frac{1}{\lambda_{0b}} \left[ 1 + \left( e^{\lambda_{0b} \tau_{\max(b)}} - 1 \right) \frac{2(y - y_w)}{P_b} \right] \qquad \frac{P_w}{2} \leq y \leq \frac{P_b}{2} + y_w \tag{4}$$

where $\tau_{\max(w)}$ and $\tau_{\max(b)}$ are the maximum shear stress at the wall and the bed of channel, respectively; $P_w$ and $P_b$ are the wetted perimeter corresponding to the wall and bed of the channel, respectively; and $y_w$ is an offset and taken as 5 mm in the analysis.

The equations presented by Knight et al. were used to predict the mean and maximum shear stress for the wall and bed of a circular flat bed channel as [52]:

$$\frac{\overline{\tau}_w}{\rho g R S} = 0.01\% SF_w (1 + P_b / P_w) \tag{5}$$

$$\frac{\overline{\tau}_b}{\rho g R S} = (1 - 0.01\% SF_w)\,(1 + P_b / P_w) \tag{6}$$

$$\frac{\tau_{\max(w)}}{\rho g R S} = 0.01\% SF_w \left[ 2.0372 (P_b / P_w)^{0.7108} \right] \tag{7}$$

$$\frac{\tau_{\max(b)}}{\rho g R S} = (1 - 0.01\% SF_w) \left[ 2.1697 (P_b / P_w)^{-0.3287} \right] \tag{8}$$

where $\overline{\tau}_w$ and $\overline{\tau}_b$ are the mean wall and bed shear stress, respectively, and $\%SF$ is the wall shear force percentage determined by the following equation [24]:

$$\%SF_w = C_{sf} \exp(-3.23 \log(P_b / C_2 P_w + 1) + 4.6052) \tag{9}$$

where $C_2 = 1.38$ and $C_{sf} = 1.0$ for $\frac{P_b}{P_w} < 4.374$, otherwise $C_{sf} = 0.6603 (P_b / P_w)^{0.28125}$.

### 2.1.2. Shannon PL Model

Sheikh and Bonakdari proposed the following equation to predict shear stress in circular channels [25]:

$$\tau = \tau_{\max} \left( \frac{y}{P/2} \right)^{1/n'} \tag{10}$$

where $n'$ is a non-dimensional parameter computed as [25]:

$$n' = \frac{\overline{\tau}}{\tau_{\max} - \overline{\tau}} \tag{11}$$

where $\overline{\tau}$ is the mean shear stress value. It should be noted that these equations are used to separately predict the wall and bed shear stresses in circular channels with a flat bed. The mean and maximum shear stress values in Equation (11) are obtained from Equations (5)–(8) and used in the entropy models presented below.

### 2.1.3. Tsallis Entropy Model

Bonakdari et al. employed the concept of Tsallis entropy to present the following relationship for estimating shear stress in a circular channel [33]:

$$\tau = \frac{k}{\lambda_1}\left[\left(\frac{\lambda_2}{k}\right)^k + \frac{\lambda_1 y}{P}\right]^{1/k} - \frac{\lambda_2}{\lambda_1} \tag{12}$$

where $k = \frac{q-1}{q}$, $q$ is a real value, and $\lambda_1$ and $\lambda_2$ are Lagrange multipliers that can be determined from the two following equations [33]:

$$[\lambda_2 + \lambda_1 \tau_{max}]^k - [\lambda_2]^k = \lambda_1 k^k \tag{13}$$

$$\tau_{max}(k+1)\lambda_1[\lambda_2 + \lambda_1 \tau_{max}]^k - [\lambda_2 + \lambda_1 \tau_{max}] = (k+1)\lambda_1^2 k^k \overline{\tau} \tag{14}$$

### 2.1.4. Renyi Model

Khozani and Bonakdari employed the Renyi entropy model to estimate the distribution of shear stress and introduced the following equation [22]:

$$\tau = \tau_{max}\left(\frac{1}{\lambda'}\left[-\lambda'' - \left((-\lambda'')^{k'} - \frac{\lambda'\alpha'^{k'}}{(\alpha'-1)}\frac{y}{P/2}\right)^{1/k'}\right]\right) \tag{15}$$

where $k' = \frac{\alpha'}{\alpha'-1}$ and $\alpha'$ is a real number between zero and one. $\lambda'$ and $\lambda''$ are Lagrange multipliers that can be calculated with two following equations [22]:

$$\frac{(-\lambda'' - \lambda')^{k'} - (-\lambda'')^{k'}}{\lambda'} = \frac{\alpha'^{k'}}{1 - \alpha'} \tag{16}$$

$$\frac{-1}{\lambda'}(-\lambda'' - \lambda')^{k'} - \frac{1}{\lambda'^2(k'+1)}\left[(-\lambda'' - \lambda')^{k'+1} - (-\lambda'')^{k'+1}\right] = \frac{\alpha'^{k'}}{(\alpha'-1)}\hat{\overline{\tau}} \tag{17}$$

where $\hat{\overline{\tau}} = \overline{\tau}/\tau_{max}$ is the dimensionless mean shear stress.

### 2.2. Global Shear Stress ($\rho g R s$)

The shear stress in open channels in case of uniform flow is considered as a basic model for comparison with entropy models as follows [53]:

$$\tau = \rho g R s \tag{18}$$

### 2.3. Data Collection

The measured shear stress data were collected from experimental results with clear and straightforward experimental conditions and included the whole range of parameters required to calculate shear stress variable [54]. Accordingly, these data series have been used by many researchers [3,24,33,55]. Sterling measured shear stress values along the wetted perimeter of a circular channel with a diameter of 244 mm in different flow conditions, as shown in Table 1 [54]. Sterling laid a thickness of sediment in a circular channel [54]. As shown in Table 1, the shear stress values are measured in four flow depths in the circular channel ($t/D = 0$) and the rest of the values related to a circular flat bed channel ($t/D \neq 0$) with different bed/sediment thickness. These measured values are considered in the uncertainty analyses of the four entropy models. In Table 1, $Q$ represents flow discharge with a unit of ($l/s$), $Fr$ is Froude number, and $S_0$ is the longitudinal slope of water surface related to experimental characteristics.

**Table 1.** Summary of the hydraulic parameters in the circular channel with and without sediment [54].

| Sample | Section | $t/D$ | $h + t/D$ | $S_0 \times 10^3$ | Fr | $Q$ (l/s) |
|---|---|---|---|---|---|---|
| 1 | | | 0.333 | 1 | 0.516 | 5.36 |
| 2 | Circular | 0 | 0.506 | 1 | 0.505 | 11.7 |
| 3 | | | 0.666 | 1 | 0.441 | 17.3 |
| 4 | | | 0.826 | 1 | 0.375 | 22.9 |
| 5 | | | 0.332 | 1.96 | 0.671 | 1.32 |
| 6 | | | 0.499 | 1.96 | 0.748 | 8 |
| 7 | | | 0.398 | 1.96 | 0.656 | 3.3 |
| 8 | | | 0.666 | 1.96 | 0.68 | 16.5 |
| 9 | Circular with flat bed | 0.25 | 0.755 | 1.96 | 0.663 | 22.1 |
| 10 | | | 0.795 | 1.96 | 0.626 | 23.8 |
| 11 | | | 0.333 | 8.62 | 1.71 | 3.39 |
| 12 | | | 0.499 | 8.62 | 1.7 | 18.2 |
| 13 | | | 0.666 | 8.62 | 1.59 | 38.9 |
| 14 | | | 0.499 | 2 | 0.718 | 4.4 |
| 15 | | | 0.666 | 2 | 0.685 | 12.2 |
| 16 | Circular with flat bed | 0.332 | 0.75 | 2 | 0.669 | 17 |
| 17 | | | 0.8 | 2 | 0.721 | 22.1 |
| 18 | | | 0.499 | 2 | 1.96 | 12 |
| 19 | | | 0.666 | 9 | 1.4 | 8.4 |
| 20 | Circular with flat bed | 0.5 | 0.75 | 9 | 1.42 | 16 |
| 21 | | | 0.8 | 9 | 1.33 | 20 |
| 22 | Circular with flat bed | 0.664 | 0.75 | 8.8 | 1.44 | 3.09 |
| 23 | | | 0.8 | 8.8 | 1.55 | 4.93 |

### 2.4. Uncertainty Analysis

The uncertainty method presented in Kazemian-Kale-Kale et al. is used to analyze the uncertainty of four entropy models of shear stress predictor [51]. This method is based on Bayesian Recursive Estimation (BaRE) algorithm and Monte-Carlo simulation [36,48]; therefore, this study is called HBMES-1 (Hybrid Bayesian and Monte-Carlo Estimation System in the first stage). The HBMES-1 uncertainty is improved and used to analyze four entropy models in shear stress prediction. This method of uncertainty is called HBMES-2.

### 2.4.1. HBMES-1 Uncertainty Method

In the BaRE algorithm, the error values between the predicted and observed values are calculated and then the appropriate transfer function is used to obtained measurements error in transformed space [8]. In the present paper, based on the BaRE algorithm proposed by Thiemann et al., in measuring the error values, the uncertainty of entropy model in predicting the shear stress distribution is evaluated based on the Monte Carlo simulation approach. All used stages in this paper are based on the different stages of Monte Carlo method: preparation, sampling (calibration and choosing the best sample size, shear stress data considered under different hydraulic conditions), initialization (choosing the function factor ($\lambda$) as to best sample size), and prediction of the output (calculating final statistical indexes of uncertainty). Therefore, based on this issue, the HBMES name is chosen for the uncertainty method proposed in this paper.

The basis of the uncertainty method presented by Kazemian-Kale-Kale et al. [50,51] was that the error distributions of the understudy models were assumed to follow a Gaussian (normal) distribution. Therefore, they discussed the normality of the error distribution. Their research showed that the shear stress data should be normalized so that the error distribution follows the Gaussian distribution. Due to the importance of normalizing the data for the uncertainty analysis, two common transfer functions of Box–Cox and Johnson were compared to evaluate the shear stress estimations models' uncertainty [51].

This comparison showed that the error distribution of normalized shear stress data using the Box–Cox function is closer to the Gaussian distribution and the uncertainty results were better and more reliable. As a result, in the present study, using the HBMES-1 method, for uncertainty analysis, the shear stress data are normalized using the Box–Cox function to follow the entropy model error distribution from the Gaussian distribution. Kazemian-Kale-Kale et al. considered the 95% confidence bound (CB) to analyze the uncertainty and performed 15 tests for uncertainty calibration based on the same CB, considering shear stress data under different hydraulic conditions. These tests were performed to select the best sample size (shear stress data considered under different hydraulic conditions).

To choose the best sample size (SS), they examined the variation of the $N_{in}$ mean and the Box–Cox function transfer factor. Due to these two statistics, selecting the best SS will be difficult, especially when the uncertainty of several models is considered. In this study, to solve this problem, the best SS is selected based on the mean value of $N_{in}$. According to [45], when $N_{in}$ is closer to 95% CB, the assumption of the Gaussian error distribution is more satisfying. As a result, only considering $N_{in}$ changes satisfies the initial condition of the Gaussian error distribution. After selecting the best SS, the best transfer factor value is used to evaluate the uncertainty [51]. The results of this evaluation are in multiple statistics of $N_{in}$, $F_P$, $F_N$, and FREE [51]. FREE is a statistical index as equal to the absolute sum of the measured data within the confidence bound ($|F_P|$) and the absolute sum of the measured data outside the confidence bound ($|F_N|$). Using these statistics determines whether each of the entropy models is sufficiently certain to predict the shear stress.

### 2.4.2. HBMES-2 Uncertainty Method

As the HBMES-1 method requires multiple statistics for concurrent evaluation, comparing the certainty of several models using this method is very difficult, and in some cases, impossible. In this study, the HBMES-1 uncertainty method is improved, and the result of uncertainty for each model is presented as a single statistic. The HBMES-2 uncertainty process has two stages: (1) the calibration step and (2) the final analysis section and the introduction of the new statistics. The calibration step of the HBMES-2 method is similar to the HBMES-1 method and is performed with the 95% CB to select the best SS [51]. One of the main advantages of the proposed HBMES-2 compared with the HBMES-1 method is that in HBMES-2, only one statistic shows the degree of uncertainty of shear stress models, whereas the HBMES-1 method requires multiple statistics. Since the main purpose of this study is to compare the uncertainty of four models in the prediction of shear stress, it is easier to compare the uncertainties using one statistic. The following three steps show the uncertainty analysis process of the HBMES-2 method.

1. Determining the $OCB_i$ and its borders;

Kazemian-Kale-Kale et al. reported that the error distribution generated by the Box–Cox transfer function follows a normal distribution and is closer to the Gaussian distribution than the Johnson transfer function [51]. The Box–Cox transfer function is a common function that has been used in many studies [56–58]. Therefore, in this paper, the Box–Cox transfer function is chosen for normalizing data error distribution.

Initially, the shear stress data are normalized using the Box–Cox transfer function and transfer factor obtained in the best SS. The Box–Cox transfer functions for shear stress data is obtained as [50]:

$$Z(\tau(y/P), \lambda) = \begin{cases} \frac{\tau^\lambda - 1}{\lambda} & if \quad \lambda \neq 0 \\ Ln\tau & if \quad \lambda = 0 \end{cases} \qquad (19)$$

Then, based on the Bayesian Recursive Estimation (BaRE) algorithm, the Gaussian error distribution is calculated as [48]:

$$\varepsilon = Z_m - Z_p \qquad (20)$$

where $\varepsilon$ is the error of data normalization, and $Z_m$ and $Z_p$ are the normalized values of measured and predicted shear stress ($\tau_m$ and $\tau_p$), respectively. Considering a given value for $OCB_i$, the following relation is applied:

$$\left(Z(y/P)\big|^{\pm}\right)_i = Z_p(y/P) + \mu_\varepsilon \pm (u/2)_i \sigma_\varepsilon \tag{21}$$

where $\mu_\varepsilon$ and $\sigma_\varepsilon$ are the mean and standard deviation of the Gaussian error distribution of the normalized shear stress data, respectively. In Equation (21), $i = 0, 1, 2, ..., n$, $n$ is a real number related to the final value of OCB, and $i$ is the number of $(u/2)$ value. In this paper, $(u/2)_i$ is the standard normal curve coefficient. The value of $i = 0$, means that an initial assumption is considered for OCB and $(u/2)$ named as $OCB_0$ and $(u/2)_0$, and $i = n$ is related to final values of OCB and $u/2$ named as $OCB_n$ and $(u/2)_n$. The value of $(Z|^{\pm})_i$ represents the effects of the statistical indexes of Gaussian error distribution ($\mu_\varepsilon$ and $\sigma_\varepsilon$) on the shear stress data predicted by the entropy model. Equation (21) yields two values for $(Z|^{\pm})_i$ based on $(u/2)_i$ value, which is obtained according to Equation (22).

Accordingly, the value of $(u/2)_i$ is related to the considered $OCB_i$ by $\psi_i$ value which is obtained as:

$$\psi_i = \frac{OCB_i/100}{2} \tag{22}$$

where $\psi_i$ is the area below one side of the normal distribution diagram. Using $\psi_i$, from the standard normal curve table, the value of the corresponding standard coefficient ($(u/2)_i$) is obtained. The main goal of Equation (22) is to obtain $\psi_i$ and therefore $(u/2)_i$ based on the $OCB_i$. Indeed, $OCB_0$ is the value of OCB for the first assumption. For example, the first assumption for OCB is $OCB_0 = 95\%$. Now, the value of $\psi_i$ according to the Equation (22) is obtained equal to $\psi_0 = 0.4750$. The $\psi_0$ value is the area below the normal distribution (Gaussian) diagram related to the first assumption of OCB. Standard normal curve table represents the area below the normal distribution diagram for different $(u/2)$ value [55]. Accordingly, the value of $(u/2)_0$ can be obtained according to the standard normal curve table. In this case, the value of $\psi_0 = 0.4750$ is found in the standard normal curve table, accordingly, the $(u/2)_0$ value is obtained equal to $(u/2)_0 = 1.9$. Then, the related precision coefficient to $(u/2)_0$ value is obtained from the column head, equal to 0.06. Finally, $(u/2)_0$ is equal to 1.96. The values of the column head represent the precision of the related $(u/2)_0$ value. Indeed, the precision of the standard normal table to obtain the $(u/2)_i$ values is equal to the two digits after the decimal $(10^{-2})$ which is represented by $\xi$ in this paper. For example, if the value of $(u/2)_0$ is equal to 1.96, then the value of $(u/2)_1$ is obtained as: $(u/2)_1 = 1.96 + 10^{-2} = 1.97$. Now, the $N_{in}$ (the percentage of measured data within the $OCB_i$) value is obtained. If the $N_{in}$ value is equal to 100, then the calculated $(u/2)_i$ is considered as $(u/2)_n$, else the next $(u/2)$ is calculated according to the previous mentioned explanations. In this paper, considering in the $OCB_n$, all points are within the CB, and therefore, the value of $N_{in}$ should always be equal to 100%.

Now, using the obtained $(u/2)_i$, the $(Z|^{\pm})_i$ is calculated using Equation (21). In the following, the $OCB_i$ borders are obtained by the Box–Cox transfer function through the following relation:

$$\left(\tau(y/P)\big|_{\pm}\right)_i = \begin{cases} \left[ (Z(y/P)|_{\pm})_i \lambda + 1 \right]^{1/\lambda} & if \quad \lambda \neq 0 \\ \exp\left[ (Z(y/P)|_{\pm})_i \right] & if \quad \lambda = 0 \end{cases} : \tag{23}$$

where $\left(\tau(y/P)\big|^{\pm}\right)_i$ are the upper and lower borders of $OCB_i$, respectively, and $\lambda$ is the Box–Cox transfer factor. After determining $OCB_i$ borders, the distance of the measured data from the borders ($dist_x$) and the sum of the distances (FREE) can be calculated. If the $dist_x$ value is positive, then the data are inside $OCB_i$ and if the $dist_x$ is negative, then the data are outside $OCB_i$. The value of $N_{in}$ (the percentage of measured data within the $OCB_i$) is obtained using $dist_x$ values.

2. Assessment of the final OCB ($OCB_n$);

As mentioned in the previous section, the first step in determining the $OCB_n$ is assuming the $OCB_0$ value. In the following, the $(u/2)_0$ value is obtained using Equation (22) and standard normal curve table. Then, the $(u/2)_i$ is calculated using the $(u/2)_0$. As considering in this paper, in the $OCB_n$, all points are within the CB, and then $N_{in}$ should always be equal to 100%. Based on obtained $(u/2)_i$ value, if the related $N_{in}$ is equal to 100, the $OCB_n$ value is calculated according to Equation (22), and the $\psi$ value. However, if $N_{in}$ is less than 100, the $OCB_i$ values considered higher, and the three Equations of (21)–(23) are performed. This should continue until $N_{in}$ is equal to 100% and the FREE values are at their lowest value (minimum FREE value). This FREE value is called $FREE_{opt}$. The flow chart in determination of $OCB_n$ is shown in Figure 2. In this figure, as stated before, $\xi$ is the precision of the standard normal table and equal to $10^{-2}$.

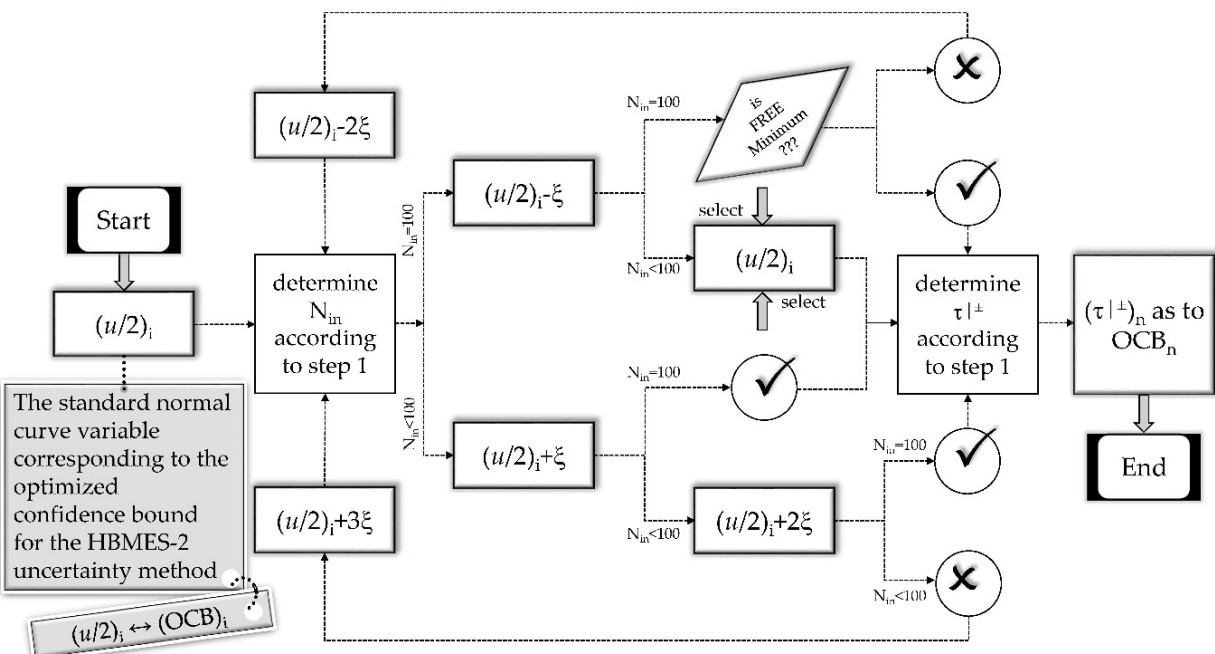

**Figure 2.** Process of assessment of the $OCB_n$ (narrowest confidence bound that covers all measured data) in HBMES-2 uncertainty method.

3. Introducing the uncertainty index of FOCB.

After determining $OCB_n$ using Equation (22), the final borders are obtained from Equation (23). The performance of the model in two ways of accuracy and precision was assessed by [36]. The authors of [36] stated that to represent the uncertainty in an efficient way, we would ideally like to have the width of prediction bounds as small as possible while containing the streamflow data. They also referred to precision as a characteristic related to the efficiency of the prediction uncertainty bounds in representing the actual distribution of the output data. Accordingly, they defined an efficiency criterion called Forecast Range Error Estimate (FREE) and introduced the FREE statistic index. FREE is a statistical index equal to the absolute sum of the measured data within the confidence bound ($|F_P|$) and the absolute sum of the measured data outside the confidence bound ($|F_N|$) (FREE = $|F_P| + |F_N|$). Moreover, Misirli et al. presented a relationship for calculating the distance of each data point from boundaries of CB called $dist_x$ based on the BaRE algorithm in estimation of maximum likelihood value of streamflow [36,49–51].

Accordingly, in this paper, for $OCB_n$, the optimized $dist_x$ value ($dist_{x(opt)}$) is optimized and derived from the following equation:

$$dist_{x(opt)} = \begin{cases} (\tau|_+)_n - \tau|_m & if \quad \tau|_m - \tau|_p \geq 0 \\ \tau|_m - (\tau|_-)_n & if \quad \tau|_m - \tau|_p < 0 \end{cases} \tag{24}$$

where $\tau_m$ and $\tau_p$ are the values of measured and predicted shear stress, respectively, and $(\tau|^{\pm})_n$ assesses the borders of $OCB_n$, which is obtained by inserting the variable of standard normal curve that relates to $OCB_n$ as $(u/2)_n$ (Equation (22)).

Moreover, based on HBMES-1, FREE evaluated the models' efficiency in the assessment of uncertainty bounds considering both the inclusion of the observed data (desirable as large as possible) and the width of the CB (to be as small as possible while maximizing the inclusion) [48]. However, in this paper, the lowest FREE value corresponding to $N_{in}$ = 100% is obtained. In HBMES-2, by considering $OCB_n$, the values of FREE are always positive, and in this paper $FREE_{opt}$ is written as follows:

$$FREE_{opt} = F_P = \sum_{dist_{x(opt)} > 0} dist_{x(opt)} \tag{25}$$

The $FREE_{opt}$ index equals the sum of the intervals of measured values inside the $OCB_n$. The $OCB_n$ and $FREE_{opt}$ statistics are quantitative and qualitative criterion, respectively, that shall be used to examine the uncertainty of the four shear stress predictor models. To consider the combined effect of these two statistics, the $FREE_{opt}$-based OCB, or the FOCB statistic, is introduced:

$$FOCB = \frac{OCB_n \times FREE_{opt}}{100} \tag{26}$$

Given that $OCB_n$ is used as a result, the present research uses the term OCB for convenience.

## 3. Results and Discussion

The results of the uncertainty calibration of the four entropy models are shown first. Then, the results of the HBMES-1 uncertainty performed at the CB = 95% and the HBMES-2 results obtained with OCB are presented.

### 3.1. Calibration

As mentioned, the calibration was executed based on $N_{in}$ changes to select the best SS. Fifteen tests were performed for calibration, and the difference in each test varied in the considered sample size (SS) (SS = 9 to SS = 23 (total 15 calibration tests). To select the best SS in each model, $N_{in}$ values of each calibration test (SS = 9 to SS = 23) were obtained at the calibration step with a default CB value which CB = 95% is considered in this paper.

Figure 3 illustrates the changes in $N_{in}$ values as a box plot for all four entropy models. The number of $N_{in}$ values in each of the calibration tests is equal to the number of SS. For example, in SS = 9, there are nine numbers for the $N_{in}$ value. Each $N_{in}$ value is related to the one shear stress data series between nine numbers of data series according to Table 1. Therefore, in each calibration test with different SS values (SS = 9 to SS = 23), there is a group of values for $N_{in}$ according to SS value. In Figure 3, each box concerning each SS value is drawn using the first and third quartiles. Moreover, the average value of $N_{in}$ between the related SS number in each box is represented by a circle form. In this figure, the red dotted line represents $N_{in}$ = 95%.

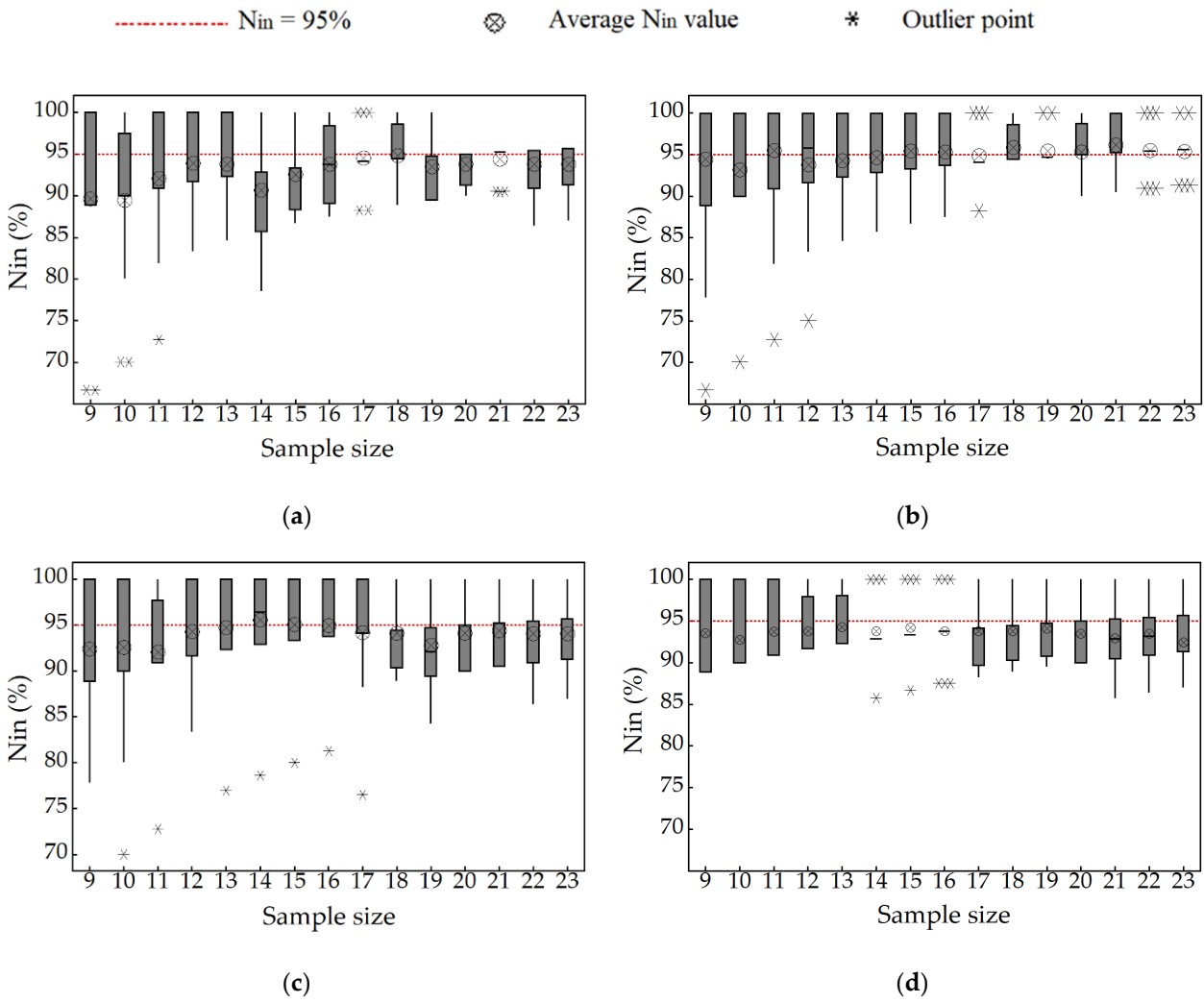

**Figure 3.** The changes in the percentage of measured values within the confidence bound ($N_{in}$) in different sample size (SS) values for all entropy models. (**a**) Tsallis entropy, (**b**) Shannon entropy, (**c**) Shannon PL entropy, and (**d**) Renyi entropy.

Each SS with an average $N_{in}$ value is closer to 95% and is selected as the best SS [49–51]. As seen in Figure 3, for the Shannon, Shannon PL, Tsallis, and Renyi entropies, the closest $N_{in}$ average values (see the circle symbol in Figure 3) to the dotted line ($N_{in}$ = 95%) are 94.85%, 95%, 94.79%, and 94.23%, respectively.

These values occur in SS amounts of 17, 15, 18, and 13, respectively. Therefore, the obtained SS values for each entropy model based on $N_{in}$ average values can be introduced as the best SS values in the calibration phase. These values of SS equal to 17, 15, 18, and 13 for Shannon, Shannon PL, Tsallis, and Renyi entropies represent at least numbers of data series that should be used for uncertainty analysis. These numbers of data series for uncertainty analysis of entropy models in predicting shear stress data are enough, and more numbers of data series are not required. Moreover, lower numbers of data series cause a risky and invalid uncertainty results. However, for further evaluation of whether or not these numbers of SS are enough, the variation of three parameters of transfer factor ($\lambda$), mean, and standard deviation of the data error distribution (i.e., $\mu_\varepsilon$ and $\sigma_\varepsilon$) are also evaluated.

After selecting the best SS, the value of the Box–Cox transfer factor ($\lambda$) in this SS is considered for the final evaluation of the uncertainty of each model. The values of transfer factor ($\lambda$) are calculated for all models in different values of SS. These values of the $\lambda$ include the average, upper, and lower limits for each SS (according to Figure 4). As can be seen in Figure 4, CB changes in all models did not significantly vary from the mean values

of $\lambda$. Thus, the mean value of $\lambda$ in the best SS (which is chosen based on $N_{in}$ values) is selected as the best $\lambda$. According to Figure 4, all model trends are similar and ascending. The changes in the CB and $\lambda$ values in the initial values of SS are high, and with increasing SS value, the $\lambda$ values decrease. The value of $\lambda$ in the optimized SS in the previous section (Figure 3) is determined as the best transfer factor ($\lambda$).

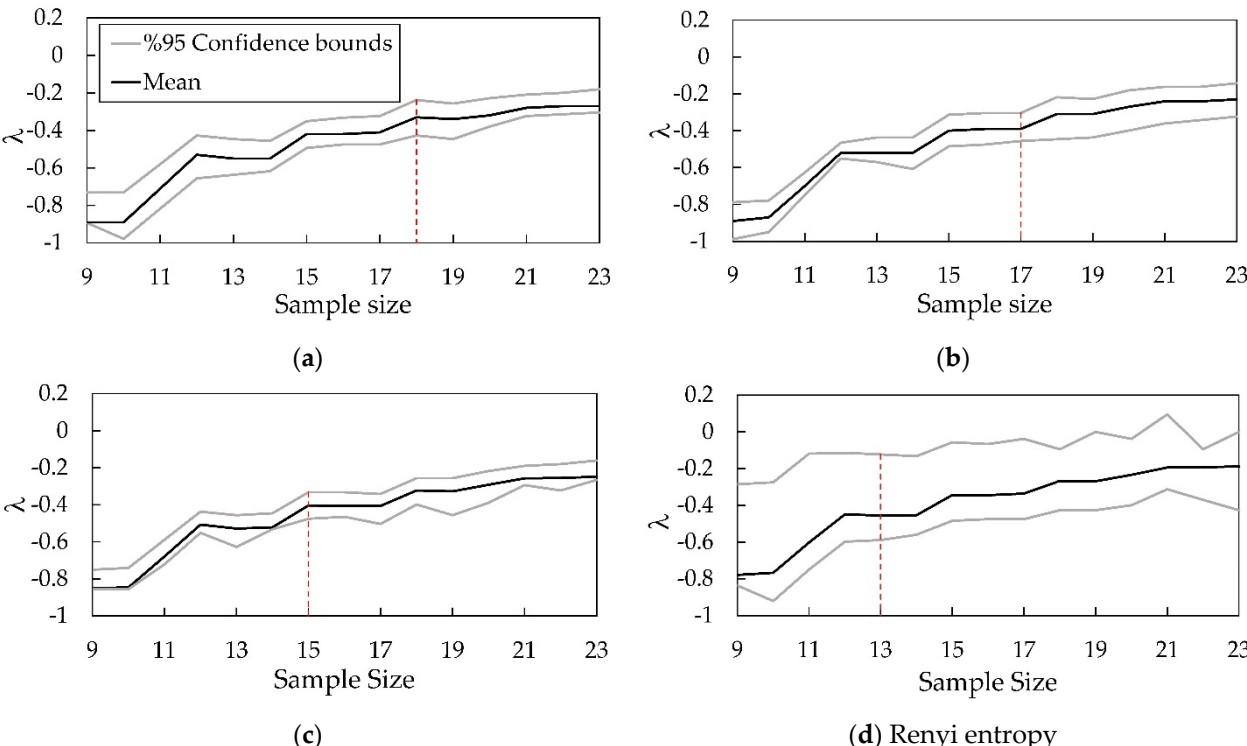

**Figure 4.** Changes in transfer factor ($\lambda$) values in different sample sizes (SSs) for all entropy models. (**a**) Tsallis entropy, (**b**) Shannon entropy, (**c**) Shannon PL entropy, and (**d**) Renyi entropy.

According to Figure 4, for all entropy models, after reaching the best SS value, the variation in the $\lambda$ is very low and tends to be constant. The constant factor $\lambda$ (zero change) after the best SS indicates that in order to achieve a Gaussian error distribution in the shear stress data series, the minimum test for each model is in the best SS value. Moreover, the notable point is the similar trend in $\lambda$ graphs and its change in the three entropies of Tsallis, Shannon, and Shannon PL, unlike the Renyi entropy model. Unlike others, in Renyi entropy, the CB variations are so significant in all SS values. Therefore, the lesser reliability of Renyi entropy is evident and clear here.

The validity of the uncertainty analysis results depends on the normality of the error distribution of each entropy model. The mean and standard deviation of the Gaussian error distribution (i.e., $\mu_\varepsilon$ and $\sigma_\varepsilon$) are also examined in the calibration stage along with the selection of the best SS. Hence, the changes in $\delta_\varepsilon$ and $\mu_\varepsilon$ were obtained for different SS values. Using the $\delta_\varepsilon$ statistic, the degree of compliance of the shear stress error distribution obtained from each entropy is determined from the Gaussian distribution. With closer error distribution to the Gaussian distribution, a greater validity of the uncertainty results is expected.

In Figure 5, the variations of $\sigma_\varepsilon$ for different SS values are illustrated for the four entropy models. As can be seen in Figure 5, for all models, the mean value of $\sigma_\varepsilon$ shows the constant trend in different SS values, and these values are 0.09, 0.06, 0.08, and 0.21 for the Shannon, Shannon PL, Tsallis, and Renyi entropies, respectively. The lower the value of $\sigma_\varepsilon$, the more the error distribution of the transferred data will follow the Gaussian distribution. Therefore, due to the small values of $\sigma_\varepsilon$ for the error distribution of the shear

stress obtained from the Shannon, Shannon PL, and Tsallis compared to Renyi entropy, it can be said that the error distribution of these three models compared to the Renyi model are closer to the Gaussian distribution. The Renyi entropy with the $\sigma_\varepsilon$ value three times larger than that of the others did not have a favorable outcome.

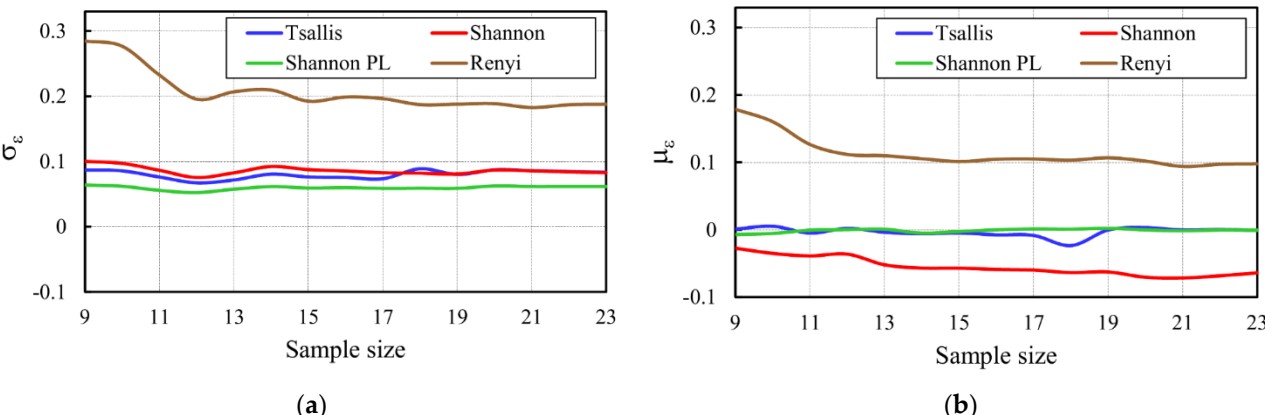

**Figure 5.** (**a**) The standard deviation and (**b**) mean value of the Gaussian error distribution ($\sigma_\varepsilon$ and $\mu_\varepsilon$) of shear stress versus 15 calibration tests with different sample sizes (SS = 9 to SS = 23) in four entropy models.

The variation in the $\mu_\varepsilon$ value is derived from all four entropy models, as illustrated in Figure 5b. As shown in this figure, the changes in $\mu_\varepsilon$ for the Renyi entropy model are higher than those of the Shannon, Shannon PL, and Tsallis models. The $\mu_\varepsilon$ values for all four models in the different SS have an almost constant value, but because the range of $\mu_\varepsilon$ variations for Renyi entropy is much higher than other models, the value of $\mu_\varepsilon$ is less reliable in this model. Therefore, since the value of $\sigma_\varepsilon$ related to error distribution of the Renyi entropy model is more than the value of $\sigma_\varepsilon$ related to other entropy models, then the error distribution of the Renyi entropy model is higher than the Gaussian error distribution. However, the mean value of the Gaussian error distribution ($\mu_\varepsilon$) is not related to how much the intended error distribution is close to the Gaussian error distribution. Sometimes, one error distribution of data series with more $\mu_\varepsilon$ value compared to other data series is close to the Gaussian error distribution. Most of the statistical theories have been presented with Gaussian error distribution [59–61].

Because the error distribution of Renyi entropy model is higher than the Gaussian error distribution, the results of the $\mu_\varepsilon$ value for Renyi entropy have lower confidence than the other entropy models. The absolute values of average $\mu_\varepsilon$ for the Shannon PL, Tsallis, Shannon, and Renyi models are 0.001, 0.003, 0.055, and 0.114, respectively. Therefore, the two entropies of Shannon PL and Tsallis contain less error than the Shannon entropy, and these three entropies perform better than Renyi entropy.

### 3.2. Assessment of Uncertainty of Four Entropy Models Using the HBMES-1 Method

By plotting the 95% CB using the HBMES-1 method, the uncertainty statistics for the four entropy models and the global shear stress model were obtained and are presented in Table 2. According to the researchers' studies, the predictor model is highly reliable if 80–100% of the values are in the desired CB, and the model was not able to predict if less than 50% of the values are within the CB [62–64]. Because of these studies, given the percentages of measured shear stress data within the CB equal to 95% ($N_{in}$), one can determine whether the entropy models are sufficiently accurate in estimating shear stress. The values of $N_{in}$ in the second column of Table 2 represent the required certainty for all models in estimating shear stress since the $N_{in}$ value for all models is greater than 85%. The $N_{in}$ values in all entropy models are very close together and higher than $N_{in}$ values for the conventional $\rho g R s$ model.

**Table 2.** Statistical indexes based on HBMES-1 uncertainty method in shear stress prediction by different entropy models and the conventional $\rho$gRs model.

|  | **Models** | $\mathbf{N_{in}}$ | $\mathbf{|F_P|}$ | $\mathbf{|F_N|}$ | **FREE** |
|---|---|---|---|---|---|
| Entropy | Shannon | 94.81 | 6.521 | 0.096 | 6.617 |
|  | Shannon PL | 93.41 | 6.018 | 0.107 | 6.125 |
|  | Tsallis | 92.43 | 8.525 | 0.239 | 8.764 |
|  | Renyi | 91.58 | 26.041 | 0.658 | 26.699 |
| Conventional | $\rho$gRs | 85.41 | 8.124 | 1.715 | 9.839 |

Although $N_{in}$ values can be used to compare each model the values of $|F_P|$ and $|F_N|$, FREE should be considered in addition to $N_{in}$ values. The $|F_P|$ values given in the third column of Table 2 represent the absolute sum of the internal data from the borders of CB. The $|F_N|$ values given in the fourth column of Table 2 represent the absolute sum of the outer data from the borders of CB. The FREE values, which are equal to the sum of $|F_P|$ and $|F_N|$, represent the WCB in the last column of Table 2. The lower values of FREE, $|F_P|$, and $|F_N|$ indicate the higher model's certainty. These three values are close for the two Shannon and Shannon PL entropies and the two Tsallis and $\rho$gRs models, but these values are much higher for the Renyi entropy than the other four models. As can be seen, the $N_{in}$ values with the three $|F_P|$, $|F_N|$, and FREE statistics give different results in providing the uncertainty, and it is not easy to give a clear view to compare the accuracy of the models.

Although the $N_{in}$ value for the Renyi entropy is more than $\rho$gRs, $|F_P|$, $|F_N|$, and FREE values are much better (lower) for the $\rho$gRs model. Therefore, the general conclusion from the values in this table is that given the high values of $N_{in}$, all models can predict shear stress with high precision, but an accurate comparison of this uncertainty concerning other statistics to be considered concurrently is complicated. In this study, HBMES-2 method is presented to solve this problem. In the next section, the results are provided and discussed in detail.

To clarify the results presented in Table 2, the shear stress distribution and 95% CB using the HBMES-1 method for a height ratio of the circular channel ($t/D = 0$, $h + t/D = 0.333$) and the height ratio of the circular channel with flat bed ($t/D = 0.25$, $h + t/D = 0.333$) is shown in Figure 6. As can be seen in Figure 6, the trend of shear stress distribution in all four entropy models is in line with the trend of measured values for the bed area and the channel walls, whereas the conventional $\rho$gRs model has both a constant amount in the bed and the walls of the channel.

In the circular channel (Figure 6a), the entropy models of Shannon, Shannon PL, and Tsallis have predicted the shear stress distribution quite following the measured data. The Renyi entropy is also in good agreement with measured data except for the channel sides ($0 < y/P < 0.1$ and $0.9 < y/P < 1$). However, the performance of the Renyi entropy is much better than the $\rho$gRs. The $\rho$gRs model at the walls of the circular channel predicts the shear stress value much lower than the measured values, and at the bed area ($0.1 < y/P < 0.9$), it predicts the shear stress value slightly below the corresponding measured values. Consequently, designing the channel based on the $\rho$gRs model, both the resistance of the walls is considered unnecessarily high, and the scouring of the bed also occurs. In the circular flat bed channel (see Figure 6b), the results of all models are similar to the circular channel. However, the Shannon PL entropy, according to the measured trend, predicts the shear stress values more than the two Shannon and Tsallis entropies.

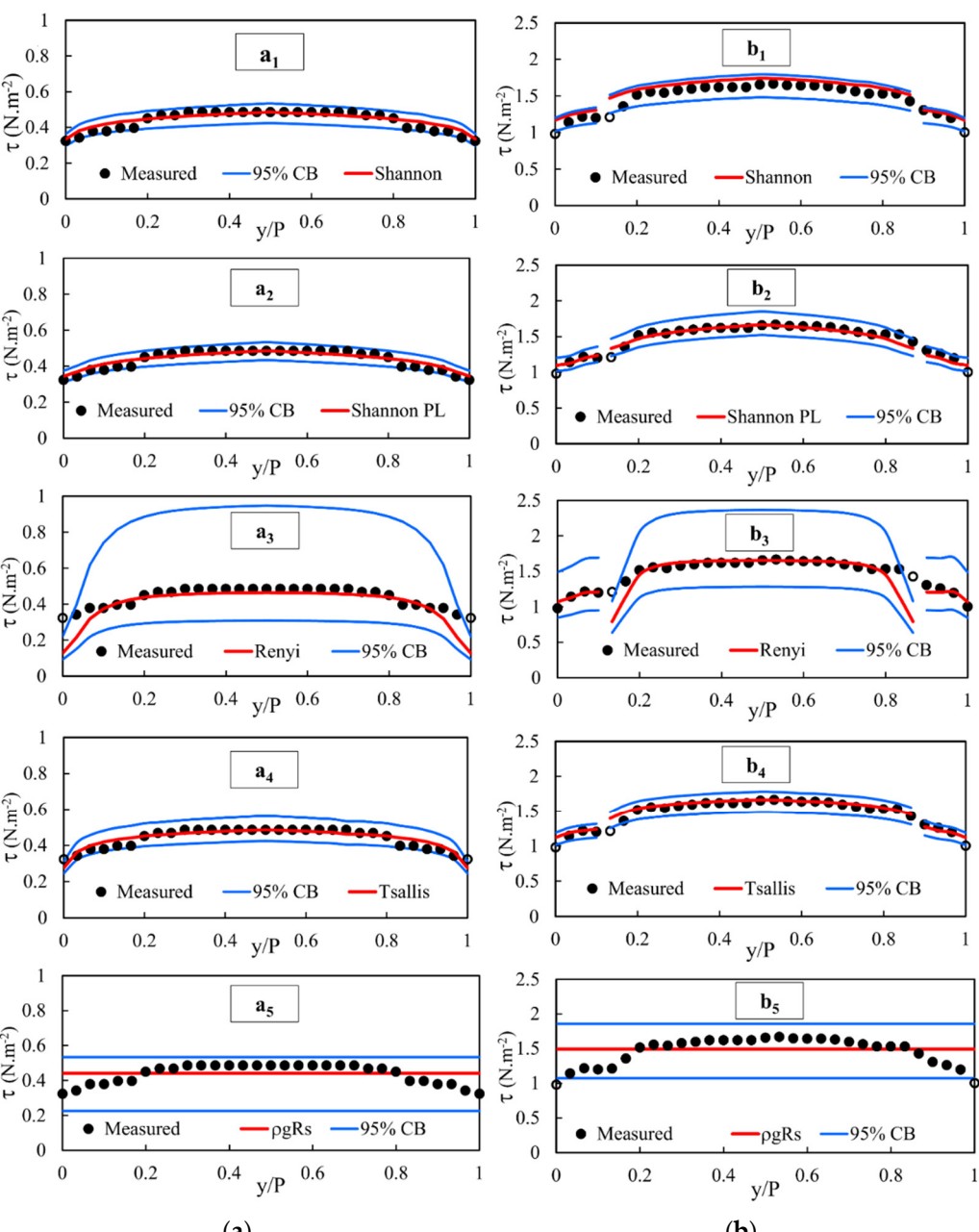

**Figure 6.** The 95% CB for uncertainty analyzing presented models in shear stress prediction by HBMES-1 method in (**a**) circular channel with height ratios $t/D = 0$, $h + t/D = 0.333$; and (**b**) circular channel with flat bed with $t/D = 0.25$; $h + t/D = 0.333$).

To check the uncertainty of the models, the WCB and the percentage of measured data within the CB ($N_{in}$) in Figure 6 should be considered. The $N_{in}$ value and the WCB at these height ratios were calculated according to the overall ability of the entropy models and considering all the 23 different hydraulic conditions in Table 1. Given that the $N_{in}$ in both height ratios for all models is more than 93%, all models can estimate shear stress with high certainty. Figure 6a shows that the CB for the Shannon and Shannon PL models with all data covered is very small and almost uniform.

In the Tsallis and Renyi models, two shear stress data in free surface ($y/P = 0$) fall outside the CB (hollow symbols in Figure 6), indicating that the certainty of these models in predicting shear stress at the free surface of the circular channel is less than the other wet areas. Such inaccuracy in prediction of shear stress can be seen in the Renyi entropy model. On the other hand, the WCB of the Tsallis model is much less than the WCB of the

Renyi model, which demonstrates more reliability in addition to the accuracy of the Tsallis model especially in areas near the water surface. The WCB of the $\rho$gRs, Shannon, and Shannon PL covered all data, but the WCB in the $\rho$gRs is larger than the other two models. Furthermore, the $\rho$gRs model cannot estimate shear stress values, so there is no consistency with the observed values. When comparing the certainty of the $\rho$gRs and Tsallis model, it should be noted that although the Tsallis entropy has two data points at the free surface outside the CB, the WCB is much lower in the Tsallis model than the $\rho$gRs model.

Therefore, Tsallis model is much more certain and accurate than the $\rho$gRs model. In addition, the presence of more coefficients in the Tsallis entropy model (two multipliers $\lambda'$, $\lambda''$, and q) can be attributed to the high accuracy of this model in the correct estimation of the shear stress value, and thus these coefficients have a significant role in predicting the shear stress distribution. In the process of solving entropy equations, the values devoted to these multipliers are adapted with the hydraulic conditions of the observational data, and the entropy models are well-adapted to provide good estimations, which is not the case with the traditional model, the $\rho$gRs method. The Renyi model also has the lowest certainty in predicting shear stress at this height ratio with the highest WCB and the least amount of $N_{in}$.

It is also observed in Figure 6b that the WCB for the Shannon, Shannon PL, and Tsallis models are approximately the same, with free surface data ($y/P = 0$) and a data point between the wall and the bed of the channel ($y/P = 0.1$) which are outside the CB. This indicates that the accuracy of these three models is lower in estimating shear stress at the free surface and the boundary of the wall and bed than other wet points. Although the Renyi entropy model has more data within the CB, its WCB is much larger than the Shannon, Shannon PL, and Tsallis models.

This issue may be related to the fact that the different performances of each model are communicated to the assumptions behind those entropy models. Accordingly, the reason for the Renyi entropy error in some areas may be the absence of dimensionality of the Lagrange multipliers within, which results in their independence of the shear stress values. It can be argued that the physical meaning effect of the Lagrange multipliers in the Renyi entropy is less than the effect of these multipliers in the Shannon and Tsallis entropies.

Comparing the uncertainty of Renyi and $\rho$gRs models, it appears that the two models have approximately the same WCB and $N_{in}$, but the certainty of the Renyi model at the intersection of the wall and bed and the certainty of the $\rho$gRs at the free surface are lower than other wet points.

### 3.3. Comparison of the Uncertainty of Four Entropy Models Using HBMES-2 Method

By drawing the 95% Confidence Bound (CB), it was found that all models had sufficient certainty in predicting shear stress in circular channels and circular flat bed channels, and the strengths and weaknesses of each model were determined. However, it is challenging and almost impossible to provide a classification and opinion on each model's final degree of uncertainty. Therefore, the HBMES-2 uncertainty method results are presented below to compare the uncertainty of the models. In this method, by drawing the narrowest confidence bound as optimized CB (OCB), all measured values are reduced within the bound. The $N_{in}$ statistic is equal to 100% and eliminated to check the uncertainty. Moreover, the FREE value introduced in previous studies, e.g., [36,49–51], is optimized (FREE$_{opt}$) and obtained according to Equations (22) and (23). The transfer factor of the Box–Cox function derived from the best SS was used to evaluate the uncertainty of the shear stress predictor models.

The introduced statistics with HBMES-2 were obtained for 23 of the samples for all entropy models, some of which are presented in Table 3. In the first column of this table, the OCB statistic shows the narrowest CB that represents all measured data. The lower the OCB value, the higher the model's certainty in the shear stress prediction. The FREE$_{opt}$ statistic, which is the total distance of the measured data of the OCB, represents the width of OCB. If this value is lower for a model, the certainty of this model is higher. As mentioned before,

the OCB criterion can be considered as a quantitative criterion for determining the certainty of a model. Moreover, $\text{FREE}_{opt}$ is a qualitative criterion to evaluate the uncertainty of models. The third column of Table 3 is the value that shows the combined effect of OCB and $\text{FREE}_{opt}$, which is considered as the main criterion in this study as (FOCB) ($\text{FREE}_{opt}$ and OCB).

**Table 3.** Statistical indexes based on HBMES-2 uncertainty method for four entropy models for shear stress prediction.

| Samples | Models | OCB | FREE$_{opt}$ | FOCB |
|---------|--------|-----|--------------|------|
| 1 | Shannon PL | 89.26 | 0.525 | 0.469 |
|   | Tsallis | 98.96 | 1.995 | 1.974 |
|   | Shannon | 92.32 | 0.718 | 0.663 |
|   | Renyi | 100 | 24.312 | 24.312 |
|   | $\rho$gRs | 87.4 | 1.628 | 1.423 |
| 2 | Shannon PL | 98.44 | 0.768 | 0.756 |
|   | Tsallis | 100 | 4.057 | 4.057 |
|   | Shannon | 95.86 | 0.975 | 0.935 |
|   | Renyi | 100 | 60.569 | 60.569 |
|   | $\rho$gRs | 93.14 | 1.48 | 1.379 |
| 8 | Shannon PL | 94.76 | 1.927 | 1.826 |
|   | Tsallis | 97.6 | 1.953 | 1.906 |
|   | Shannon | 96.06 | 2.931 | 2.815 |
|   | Renyi | 99.02 | 29.049 | 28.764 |
|   | $\rho$gRs | 99.14 | 4.92 | 4.878 |
| 11 | Shannon PL | 99.5 | 5.166 | 5.14 |
|   | Tsallis | 100 | 6.723 | 6.723 |
|   | Shannon | 99.62 | 5.727 | 5.705 |
|   | Renyi | 100 | 22.773 | 22.773 |
|   | $\rho$gRs | 98.98 | 11.157 | 11.043 |
| 18 | Shannon PL | 100 | 17.426 | 17.426 |
|   | Tsallis | 100 | 22.231 | 22.231 |
|   | Shannon | 100 | 16.049 | 16.049 |
|   | Renyi | 100 | 41.814 | 41.814 |
|   | $\rho$gRs | 99.56 | 21.233 | 21.139 |

Therefore, it is sufficient to find only the FOCB statistic for evaluating the uncertainty of shear stress predictor models. As can be seen in the last column of Table 3, the overall uncertainties of all models decrease with an increasing amount of water and sediment in the channel bed ($h + t$). In all samples, the FOCB statistic has the lowest value for the Shannon PL entropy and the highest value for the Renyi entropy, and as a result, among the proposed models, Shannon PL models showed the highest certainty, and the Renyi has the lowest certainty. The FOCB value for the Renyi entropy is more than several times that of the other models, and this indicates a much lower certainty of this model than the other models in shear stress prediction. In the three samples 1, 2, and 11, the entropies of Shannon, $\rho$gRs, and Tsallis have the lowest amount of the FOCB and the highest certainty, respectively. In samples 8 and 11, the values of the FOCB for the three entropies Shannon, $\rho$gRs, and Tsallis are also close together, indicating an almost identical degree of certainty for these models.

To illustrate the uncertainty statistics presented in Table 3, the OCB for two samples 1 and 11 is shown in Figure 7. Figure 7 shows the narrowest confidence bound that covers all measured shear stress, OCB, for the four entropy models and the $\rho$gRs model. Figure 7a corresponds to sample 1 for a circular channel ($t/D = 0$, $h + t/D = 0.333$) and Figure 7b also relates to sample 11 for a circular flat bed channel ($t/D = 0.25$, $h + t/D = 0.333$). Each model, which has a higher OCB width, shows higher uncertainty in the prediction of shear stress. The width of OCB of the three entropy models of Tsallis, Shannon PL, and Shannon

is very close and contains a very small region, while the Renyi entropy model has larger OCB region compared with other three models.

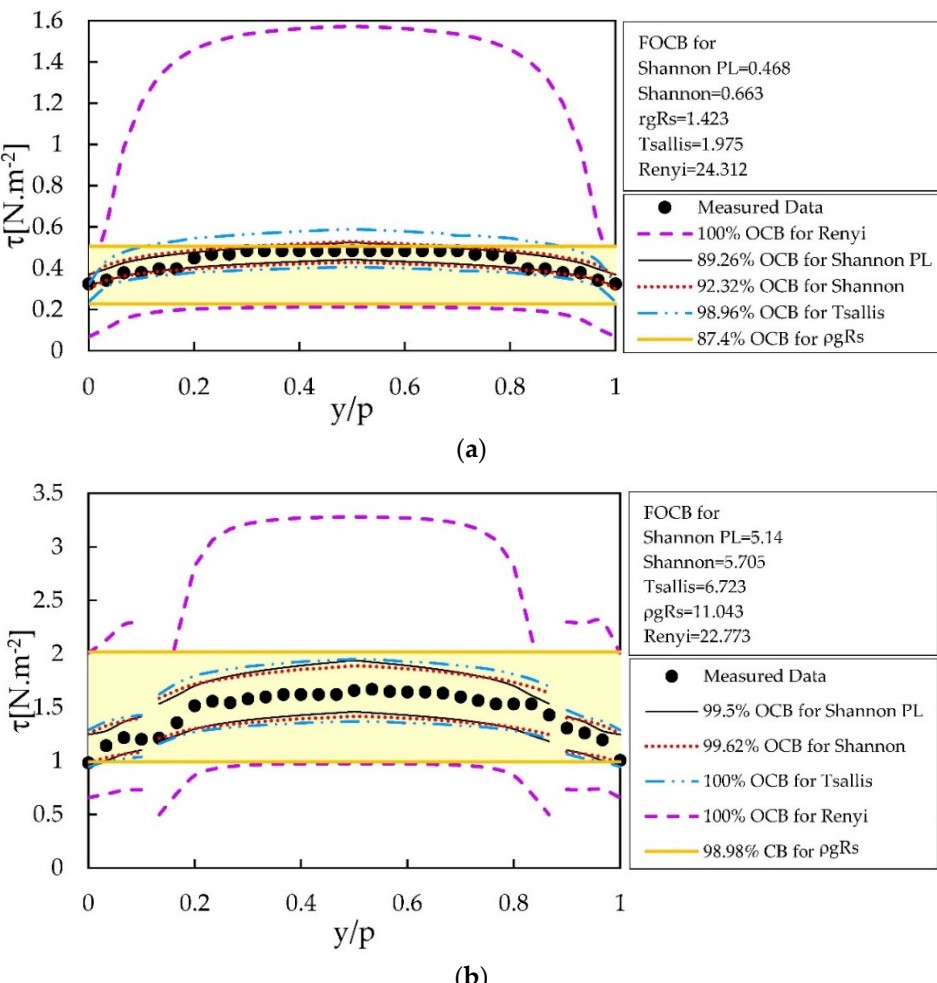

(a)

(b)

**Figure 7.** OCB for analyzing presented models in shear stress prediction by HBMES-2 uncertainty method in height ratios (**a**) $t/D = 0$, $h + t/D = 0.333$; and (**b**) $t/D = 0.25$; $h + t/D = 0.333$).

In Figure 7a, the OCB widths for the two Shannon PL and Shannon models are perfectly within the range of the OCB plotted for the $\rho$gRs model, but the OCB plotted for the Tsallis entropy is slightly wider than the OCB for the $\rho$gRs model. Therefore, the certainty of the Shannon PL, Shannon, $\rho$gRs, Tsallis, and Renyi, in order, is higher in predicting shear stress in this sample. The results of the FOCB values also confirm this classification.

In Table 3, sample 1 ($t/D = 0$), the OCB values for three Shannon PL, Shannon, and $\rho$gRs models are close to each other and are less than the OCB values of the Tsallis and Renyi models. Despite the proximity of the OCB values to the two Tsallis and Renyi models, the width of OCB for the Renyi model is much higher than the Tsallis model, and the FREE$_{opt}$ value of the Renyi model is much larger than the other four models, which is shown in Table 3. In Figure 7b, although the OCB values are very close to each other for all models, the width of OCB in the bed and the wall of the channel for the Renyi entropy model is much larger than that of the four models. This issue is very clear in the (FOCB) statistics in Table 3 for all samples.

In Table 4, the values of this statistic are given for all models for the studied cases. These values show that the certainty of all models in calculating shear stress in a circular channel is much higher than in a circular flat bed channel. Except for the Renyi entropy, with almost identical values for FOCB, it indicates the same uncertainty in the prediction

of shear stress values. In Table 4, the values of FOCB for the Renyi entropy in both circular and circular flat bed channels are much higher than in the other models. Given that the $\rho$gRs model has been considered as a basic model in this study for comparison with entropy models, it should be said that the Renyi entropy model is not a good model for predicting shear stress values. In a circular channel, the entropies of Shannon PL, $\rho$gRs, Shannon, and Tsallis have more certainty in predicting shear stress.

**Table 4.** The values of FOCB in circular and circular flat bed channels for all entropy models to predicting shear stress distribution.

| Section | FOCB | | | | |
|---|---|---|---|---|---|
| | Shannon PL | Shannon | Tsallis | Renyi | $\rho$gRs |
| Circular | 1.339 | 2.432 | 2.961 | 58.457 | 2.026 |
| Circular with flat bed | 11.591 | 10.118 | 17.407 | 57.569 | 17.115 |
| Average | 9.808 | 8.781 | 14.895 | 57.726 | 14.491 |

The models are more certain for the Shannon, Shannon PL, $\rho$gRs, and Tsallis models in predicting shear stress in circular flat bed channels. Finally, considering the number of samples presented in Table 1, the average values in Table 4 are given for the entire circular channel. The average values indicate that the Shannon, Shannon PL, $\rho$gRs, and Tsallis models have the most certainty in estimating shear stress in the circular channels. The FOCB values in all the entropy models indicate that the certainty of all models in shear stress estimation in circular channels is more than for a circular channel with a flat bed. However, the high and almost equal values of FOCB in the Renyi entropy model for circular channels and circular channels with a flat bed illustrate that the Renyi entropy model has more uncertainty than other models in both circular channels and circular channels with a flat bed.

## 4. Conclusions

In this study, the uncertainty of four popular entropy models, Shannon, Shannon PL, Tsallis, and Renyi, were analyzed for calculating shear stress in circular channels. The uncertainty analysis method based on the Bayesian Monte-Carlo technique [51] was employed and named in this study as HBMES-1. However, using the HBMES-1 method required four statistics (Nin, |FP|, |FN|, and FREE), it was not feasible to compare the uncertainty of several different entropy models.

For this reason, in the new HBMES-2 method proposed in this paper, the narrowest CB that covers all measured data, the Optimized Confidence Bound (OCB), was obtained, and a new statistic called FOCB was introduced to evaluate the uncertainty. In this study, a new statistical index called $FREE_{opt}$-based OCB (FOCB) was introduced. One of the main advantages of the proposed HBMES-2 compared with the HBMES-1 method is that in HBMES-2, only one statistic shows the degree of uncertainty of shear stress models, whereas the HBMES-1 method requires multiple statistics. The FOCB indicated the degree of uncertainty. In the calibration stage for both uncertainty methods, based on the percentage of measured data within the confidence bound (Nin), the best SS (sample size) for each entropy model was selected.

At the calibration stage, based on the obtained $\lambda$ (the Box–Cox function transfer factor) value in the best SS, the final evaluation was performed using two uncertainty methods of HBMES-1 and HBMES-2. In the HBMES-1 method, it was found that all four entropy models, along with the $\rho$gRs conventional model, with the $N_{in}$ values higher than 93% have high certainty in predicting shear stress in circular channels.

According to the results of the HBMES-2 method, in a circular channel, the entropy models of Shannon PL, $\rho$gRs, Shannon, Tsallis, and Renyi with the lowest FOCB values equal to 1.339, 2.026, 2.432, 2.961, and 58.457, respectively, had the highest certainty with the FOCB values. Furthermore, in the circular flat bed channel, the entropy models of Shannon

PL, Shannon, ρgRs, Tsallis, and Renyi, had the lowest uncertainty with the FOCB values equal to 10.118, 11.591, 17.115, 17.407, and 57.565, respectively. Based on the mean results of FOCB in circular and circular flat bed channels, it was generally found that the Shannon PL, Shannon, ρgRs, Tsallis, and Renyi models had the highest certainty in shear stress prediction with FOCB values equal to 8.781, 9.808, 14.491, 14.895, and 57.726, respectively. These results showed that the Shannon, Shannon PL, and Tsallis entropy models, along with the ρgRs conventional model, had the lowest uncertainty in shear stress prediction, whereas the Renyi entropy model had the highest uncertainty in predicting shear stress values in the circular channels. However, more research is needed to investigate the uncertainty of these four entropy models with the proposed HBMES-2 method in different cross-section channels.

**Author Contributions:** Conceptualization, A.K.-K.-K. and H.B.; methodology, A.K.-K.-K. and A.G.; software, A.K.-K.-K.; formal analysis, A.K.-K.-K., A.G. and H.B.; investigation, A.K.-K.-K., A.G. and H.B.; resources, A.K.-K.-K. and A.G.; data curation, A.K.-K.-K. and A.G.; writing—original draft preparation, A.K.-K.-K., A.G., M.R.-B., A.M., A.A.S., A.H.A., B.G. and H.B.; writing—review and editing, A.K.-K.-K., A.G., A.H.A., B.G. and H.B.; visualization, A.K.-K.-K., A.G. and H.B.; supervision, H.B.; project administration, A.H.A., B.G. and H.B. All authors have read and agreed to the published version of the manuscript.

**Funding:** This research received no external funding.

**Institutional Review Board Statement:** Not applicable.

**Informed Consent Statement:** Not applicable.

**Data Availability Statement:** All data, models, and code generated or used during the study appear in the published article.

**Conflicts of Interest:** The authors declare no conflict of interest.

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
