# Peer review of "Uncertainty Assessment of Entropy-Based Circular Channel Shear Stress Prediction Models Using a Novel Method"

_geosciences, doi:10.3390/geosciences11080308_

Round 1
Reviewer 1 Report
Please find the attachment.

Author Response
Reviewer: 1
General Comments:
The present study proposes a method to evaluate the uncertainty of four entropy models, namely, Shannon, Shannon-Power Law (PL), Tsallis, and Renyi, in the context of shear stress prediction for circular channels. Based on the new method, the authors have used the effect of Optimized Forecasting Range of Error Estimation (FREEopt) and Optimized Confidence Bound (OCB) to introduce a new statistic index, called FREEopt-based OCB (FOCB). The lower value FOCB value corresponds to the more certainty of the model. Then, the analysis has been carried out.
The authors have carried out a detail data analysis in the manuscript. The corresponding figures and tables are adequately presented and discussed properly. Therefore, I recommend the publication of the paper subject to some minor revisions given below.
Response: Thank you. We greatly appreciate the reviewer’s efforts to review the paper and provide their valuable suggestions. We have addressed all issues indicated in the review report. The changes are tracked in the manuscript to show all the changes that are done in response to the reviewer’s comments.
Specific Comments:
- Line 21: Please correct Shannon-Power Low (PL).
Response: (Please see “Abstract”, “Keywords”)
- Lines 50-51: Please rephrase the sentence ‘Another entropies...’.
Response: Tsallis and Renyi entropies were applied for characterizing threshold channel bank profiles and distribution of sediment concentration, respectively. (Please see P. 2)
- Line 53: What do you mean by ‘non-additive parameters’?
Response: The parameters used in Tsallis entropy are the same as those used in Shannon entropy. However, the Tsallis entropy is less sensitive to the shape of the probability density function (PDF). In other words, Tsallis entropy for different PDF is more responsive than Shannon entropy. (Please see P. 2)
- Similar to the above 3 comments, the paper contains several grammatical errors, and some of the sentences are difficult to follow. Therefore, I suggest the authors to recheck them.
Response: In addition to the above comments, we have carefully proofread the manuscript, and any grammatical errors have been corrected.
We hope these changes have addressed all of the Reviewer's comments regarding the work.
Reviewer 2 Report
The Authors should consider my comments in the attached file.

Author Response
Reviewer: 2
-For a recent study about the application of Shannon entropy to velocity distribution (in a canopy flow), see:
- Mihailović, G. Mimić, P. Gualtieri, I. Arsenić, C. Gualtieri, Randomness representation of turbulence in canopy flows using Kolmogorov complexity measures, Entropy 19 (2017) 519, http://dx.doi.org/10.3390/e19100519
- Mihailović, G. Mimić, P. Gualtieri, I. Arsenić, C. Gualtieri, (2016). Randomness representation in turbulent flows with bed roughness elements using the spectrum of the Kolmogorov complexity. Environmental Modelling and Software for Sustainability in a Context of Global Change.
Response: Thank you very much for your comments. The novel application of the entropy theory has been successful in modelling: velocity distribution [17,18,19], the transverse slope of stable channels banks [20,21], and shear stress distribution [22].
References
- Mihaliović, D. T.; Mimić, G.; Gualtieri, P.; Arsenić, I.; Gualtieri, C. Randomness representation in turbulent flows with bed roughness elements using the spectrum of the Kolmogorov complexity. In Environmental Modeling and Software for Sustainability in a Context of Global Change, Proceedings of the iEMSs Eight Biennial Meeting: International Congress on Environmental Modeling and Software (iEMSs), Toulouse, France, July, 10/14, 2016
- Mihaliović, D. T.; Mimić, G.; Gualtieri, P.; Arsenić, I.; Gualtieri, C. Randomness Representation of Turbulence in Canopy Flows Using Kolmogorov Complexity Meaures. Entropy 2017, 19, 10, 519. https://doi.org/10.3390/e19100519
(Please see “Introduction”, and “References”)
-The text from line 63 to line 89 and from line 105 to line 121 should be shortened.
Response: Thank you very much for your comments. The uncertainty analysis of many hydraulic and hydrological models has been tackled by many researchers [34-47]. Thiemann et al. developed a Bayesian formulation, which permits the hydrologist to quantify uncertainty [48]. The method is called Bayesian Recursive Estimation (BaRE). Misirli et al. using BaRE and Monte-Carlo simulation (BMC) presented an uncertainty method to analyze stream flow uncertainty [36]. Then uncertainty method of Misirli et al., applied to the uncertainty analysis of Shannon entropy model in estimating the velocity distribution [49]. Kazemian-Kale-Kale et al. with the improving uncertainty method used in previous studies [36,48,49] analyzed uncertainty of the Tsallis entropy model in predicting shear stress distribution [50]. They emphasized on the necessity to normalize the shear stress data for the uncertainty analyses. Then, they calibrated the model to select the best sample size (shear stress data considered under different hydraulic conditions) and finally analyzed the uncertainty of the Tsallis entropy model using this sample size (SS). Although their results were well capable of analyzing the uncertainty of the Tsallis model, their calibration method was difficult. They did not discuss the performance of the different transfer functions to normalization in the uncertainty results. Therefore, in another study, Kazemian-Kale-Kale et al. simplified the calibration method of their previous study, and in addition to the Box-Cox function, Johnson’s function was used to analyze the Shannon entropy uncertainty [51].
However, it is difficult to compare the certainty of five shear stress prediction models. Therefore, the HBMES-1 method is further improved in this study, and the uncertainty method is presented as HBMES-2. The Minimum CB that covers all measured data; OCB (Optimized Confidence Bound) is defined by employing the HBMES-2 method. Then, Based on the OCB, the FREE statistic is optimized and is called FREEopt. Given the value of OCB and FREEopt, the FOCB statistic is introduced that can show the effect of all uncertainty statistics. The drawn OCB represents the width of the confidence bound, which is a quantitative statistic for estimating uncertainty. In addition, FREEopt, which represents the absolute sum of the measured data within the OCB, is a qualitative statistic for estimating uncertainty. FOCB shows the degree of qualitative and quantitative uncertainty of shear stress models. Therefore, it is easy to compare using only one statistic (FOCB) in the HBMES-2 method.
(Please see “Introduction”)
-This comparative analysis is not enough. The Authors should explain why these models have different performances. In other words, the Author should discuss whether and how the different performance of each model are related to the assumptions behind those models.
Response: Thank you very much for your comments. The entropy theory was developed by Shannon (1948) to express information or uncertainty in the field of communication, which is considered as a beneficial characteristic of any probability density function (PDF). Renyi entropy is defined as a generalized form of entropy that was introduced by Renyi (1961). Renyi entropy can be considered as a generalization of the Shannon Entropy (Jozsa, 1998; Titchener et al., 2004). Tsallis (1988) introduced a generalized form of entropy. Tsallis entropy is express as a generalization of the Shannon entropy or Boltzmann-Gibbs entropy.
Another form of Shannon entropy was proposed by Khozani and Bonakdari (2018) to estimate shear stress and called Shannon PL entropy. The extracted equations for calculating shear stress using these entropies are all obtained using the principle of maximum entropy (POME) (Jaynes, 1957) and two constraints. Shannon and Tsallis entropies calculations and how these constraints are applied to the shear stress distribution are presented by Bonakdari et al. (2015). Given that the constraints are the same in all entropies presented in this study, the difference is in the initial form of the introduction of entropy. Using the POME and considering the constraints in the equation presented using Shannon entropy, we have one Langrage multiplier (equation 2 in the manuscript file). Still, we have two Langrage multipliers in the Tsallis and Renyi entropies (equations 12 and 15, respectively). Therefore, it can be said that the main reason for the difference in entropies in estimating shear stress is their dependence on Langrage multipliers and their sensitivity to these multipliers. The importance of the effect of Langrage multipliers in the study of sterling and Knight (2002) is mentioned as follows:
“However, it has to be said that based on the entropy approach, the results are generally disappointing. This paper is therefore presented to stimulate discussion on this issue. Maybe there is some value in the Langrage multiplier λ, or indeed entropy. It is hoped that hydraulic engineers will be stimulated to consider this approach afresh. It is clear that entropy requires further study before it can be widely applied in hydraulic engineering with confidence.” In the process of solving entropy equations, the values devoted to these multipliers are adapted with the hydraulic conditions of the observational data, and the entropy models are well-adapted to provide good estimations, which is not the case with the traditional model such ρgRs method.
Also, MasZczyk and Dush 2008 stated that having a more Langrage multiplier in Tsallis entropy than Shannon entropy made Tsallis entropy less sensitive to the PDF. The extraction calculations of Langrage multipliers and their different values on results of shear stress distribution in the study of Bonakdari et al. (2015) are given.
Due to the sensitivity of entropy equations to Langrage multipliers, Khozani and Bonakdari 2018 presented a relation using Shannon entropy. The Langrage multipliers were removed and called Shannon Power Law (PL) entropy. Their study showed that the Shannon PL entropy error in estimating shear stress is lower than other entropies.
Moreover, the reason for the Renyi entropy’s error in some areas may be the absence of dimensionality of the Lagrange multipliers within, which results in their independence of the shear stress values. It can be argued that the physical meaning effect of the Lagrange multipliers in the Renyi entropy is less than the effect of these multipliers in the Shannon and Tsallis entropies (Cao and Knight 1997; Singh and Luo 2011). (Please see section “3.2”)
As mentioned, many studies have been done on how to accurately calculate the different types of entropies, the inputs required to calculate the shear stress and the Lagrangian coefficients. Despite these extensive studies, the question arises as to whether entropy models are able to estimate shear stress in practical projects? Which entropy model is more accurate in estimating shear stress than other entropies? therefore, the uncertainty analysis of the results of these models can be a bridge between these studies and the selection of the most reliable entropy model for estimating shear stress. For this reason, in this study, only the results of entropy models are analyzed, and the focus of this study is not on the initial parameters and differences of the initial assumptions in entropy models. In this study, the uncertainty of four entropy models in estimating shear stress in circular and circular flatbed channels under the same conditions (table 1 in the manuscript file) is evaluated using a new method (HBMES-2). In HBMES, the outputs of entropy models are analyzed, not the inputs of entropy models.
References
Shannon, C. E. (1948). “A mathematical theory of communications, I and II.” Bell Syst. Tech. J., 27, 379-423.
Renyi, A. (1961). “On measures of entropy and information. Proceedings, 4th BerkeleySymposium on Mathematics.” Stat. Probabil., 1, 547-561.
Jozsa, R. (1998). “Quantum Information and Its Properties.” In Introduction to Quantum Computation and Information; Lo, H.K., Popescu, S., Spiller, T., Eds.; World Scientific: Singapore.
Titchener, M. R., Nicolescu, R., Staiger, L., Gulliver, A., and Speidel, U. (2004). “Deterministic Complexity and Entropy.” J. Fundam. Inf. 64(1-4), 443-461
Tsallis, C. (1988). “Possible generalization of Boltzmann–Gibbs statistics.” J. Stat. Phys., 52(1-2), 479-487.
Jaynes, E. T. (1957a). “Information theory and statistical mechanics 1.” Phys. Rev., 106(4), 620-630.
Bonakdari, H.; Sheikh, Z.; Tooshmalani. M. Comparison between Shannon and Tsallis entropies for prediction of shear stress distribution in open channels. Stochastic Hydrol. Hydraul. 2015, 29, 1–11. https://doi.org/10.1007/s00477-014-0959-3
Sterling, M., and Knight, D. W. (2002). “An attempt at using the entropy approach to predict the transverse distribution of boundary shear stress in open channel flow.” J. Stochastic Environ. Res. Risk Assess. 16, 127-142.
Khozani, Z. S.; Bonakdari, H. Formulating the shear stress distribution in circular open channels based on the Renyi entropy. Physica A 2018, 490, 114–126. https://doi.org/10.1007/s11709-020-0634-3.
Cao S, Knight DW (1996) Shannon’s entropy-based bank profile equation of threshold channels. Stochastic Hydraulics ’96, Proceedings of the seventh IHAR international symposium, Mackay, Queensland, Australia.
Singh, V. P., & Luo, H. (2011). Entropy theory for distribution of one-dimensional velocity in open channels. Journal of Hydrologic Engineering, 16(9), 725-735.
-The Authors should try to explain why Renyi method performs differently from the other methods. It is expected that the above result holds even for a cross-section having a different shape (not circular) ?
Response: Thank you very much for your comments. The Renyi entropy’s error in some areas may be the absence of dimensionality of the Lagrange multipliers within, which results in their independence of the shear stress values. It can be argued that the physical meaning effect of the Lagrange multipliers in the Renyi entropy is less than the effect of these multipliers in the Shannon and Tsallis entropies (Cao and Knight 1997; Singh and Luo 2011). The results of this study are verified for circular and circular with flatbed channels and hydraulic parameters as presented in Table 1. However, using the HBMES-2 uncertainty method, the uncertainty of Renyi entropy can be evaluated for other cross-section having a different shape. (Please see section “3.3”)
References:
Cao S, Knight DW (1996) Shannon’s entropy-based bank profile equation of threshold channels. Stochastic Hydraulics ’96, Proceedings of the seventh IHAR international symposium, Mackay, Queensland, Australia.
Singh, V. P., & Luo, H. (2011). Entropy theory for distribution of one-dimensional velocity in open channels. Journal of Hydrologic Engineering, 16(9), 725-735.
-Table 4 show a large difference in FOCB between circular and circular with flatbed channels for Shannon PL. Shannon and Tsallis, but not for Renyi. The Authors should comment on this.
Response: Thank you very much for your comments. FOCB values in all models indicate that the certainty of all models in estimating shear stress in circular channels is much higher than circular with flatbed channels. But Large and relatively equal values of FOCB for Renyi entropy in circular and circular with flatbed channels show that this model has more uncertainty than other models in circular and circular with flatbed channels..
(Please see “Comparison of the Uncertainty of Four Entropy Models Using HBMES-2 Method”)
-This Section should be drastically shortened pointing out only the major findings from this study.
Response: Thank you very much for your comments. In this study, the uncertainty of four popular entropy models, including Shannon, Shannon PL, Tsallis, and Renyi were analyzed for calculating shear stress in circular channels. The uncertainty analysis method based on the Bayesian Monte-Carlo technique [51] was employed and named in this study as HBMES-1. However, using the HBMES-1 method required four statistics (Nin, |FP|, |FN|, and FREE), it was not feasible to compare the uncertainty of several and different entropy models.
For this reason, in the new HBMES-2 method proposed in this paper, the narrowest CB that covers all measured data; Optimized Confidence Bound (OCB); was obtained, and a new statistic called FOCB was introduced to evaluate the uncertainty. The FOCB indicated the degree of uncertainty. In the calibration stage for both uncertainty methods, based on the percentage of measured data within the confidence bound (Nin), the best SS (sample size) for each entropy model was selected.
At the calibration stage, based on the obtained λ (the Box-Cox function transfer factor) value in the best SS, the final evaluation was performed using two uncertainty methods of HBMES-1 and HBMES-2. In the HBMES-1 method, it was found that all four entropy models, along with the rgRs conventional model with the Nin values higher than 93% have high certainty in predicting shear stress in circular channels.
According to the results of HBMES-2 method in a circular channel, the entropy models of Shannon PL, rgRs, Shannon, Tsallis, and Renyi with the lowest FOCB values equal to 1.339, 2.026, 2.432, 2.961, and 58.457, respectively had the highest certainty with the FOCB values. Furthermore, in the circular with flatbed channel, the entropy models of Shannon PL, Shannon, rgRs, Tsallis, and Renyi, had the lowest uncertainty with the FOCB values equal to 10.118, 11.591, 17.115, 17.407, and 57.565, respectively. Based on the mean results of FOCB in circular and circular flatbed channels, it was generally found that the Shannon PL, Shannon, rgRs, Tsallis, and Renyi models had the highest certainty in shear stress prediction with FOCB values equal to 8.781, 9.808, 14.491, 14.895, and 57.726, respectively. These results showed that the three Shannon, Shannon PL, and Tsallis entropy models, along with the rgRs conventional model, had the lowest uncertainty in shear stress prediction, whereas the Renyi entropy model had the highest uncertainty in predicting shear stress values in the circular channels.
Moreover, the result of this study can not be generalized to other cross-section channel as well as other models for estimating shear stress. In addition, the results of this study cannot be generalized to other channels with diferent shape as well as other models for estimating shear stress. Therefore, researchers interested in this field are recommended to use the HBMES-2 method to evaluate the uncertainty of entropy models and classical method based upon momentum equation in estimating shear stress in diferent cross-section channels (rectangular, trapezoidal and etc).
(Please see “Conclusion”)
I think that the manuscript suffers from two issues:
- The comparative analysis of the tested methods is weak. The Authors compare their perfomances but they dont provide any explanation for the differences among those performances. In other words, the Author should discuss whether and how the different performance of each model are related to the assumptions behind those models. At the end, one could ask: which method should be preferred in practical cases ? Are the conclusions presented in this study applicable to a channel cross-section having a different shape (not circular) ? Why should those entropic methods be preferred to the classical method based upon momentum equation ?
Response: Thank you very much for your comments. Each of the introduced entropies, which are all a type of Shannon entropy, used by a number of researchers to obtain shear stress. In addition, another form of Shannon entropy was proposed by Khozani and Bonakdari (2018) to estimate shear stress and called Shannon PL entropy. The extracted equations for estimating shear stress using these entropies are all obtained using the principle of maximum entropy (POME) (Jaynes, 1957) and two constraints. Shannon and Tsallis entropies calculations and how these constraints are applied to the shear stress distribution are presented by Bonakdari et al. (2015). Using the POME and considering the constraints in the equation presented using Shannon entropy, we have one Langrage multiplier (equation 2 in manuscript file), but we have two Langrage multipliers in the Tsallis and Renyi entropies (equations 12 and 15, respectively). Therefore, it can be said that the main reason for the difference in entropies in estimating shear stress is their dependence on Langrage multipliers and their sensitivity to these multipliers. The importance of the effect of Langrage multipliers in the study of sterling and Knight (2002) is mentioned as follows:
“However, it has to be said that based on the entropy approach the results are generally disappointing. This paper is therefore presented in order to stimulate discussion on this issue. Maybe there is some value in the Langrage multiplier λ, or indeed entropy. It is hoped that hydraulic engineers will be stimulated to consider this approach afresh. It is clear that entropy requires further study before it can be widely applied in hydraulic engineering with confidence.”. In the process of solving entropy equations, the values devoted to these multipliers are adapted with the hydraulic conditions of the observational data, and the entropy models are well-adapted to provide good estimations, which is not the case with the traditional model such ρgRs method.
Also, MasZczyk and Dush 2008 stated that having a more Langrage multiplier in Tsallis entropy than Shannon entropy made Tsallis entropy less sensitive to the PDF. The extraction calculations of Langrage multipliers and their different values on results of shear stress distribution in the study of Bonakdari et al. (2015) are given.
Due to the sensitivity of entropy equations to Langrage multipliers, Khozani and Bonakdari 2018 presented a relation using Shannon entropy. The Langrage multipliers were removed and called Shannon Power Law (PL) entropy. The results of their study showed that the Shannon PL entropy error in estimating shear stress is lower than other entropies.
Morover, the reason for the Renyi entropy’s error in some areas may be the absence of dimensionality of the Lagrange multipliers within, which results in their independence of the shear stress values. It can be argued that the physical meaning effect of the Lagrange multipliers in the Renyi entropy is less than the effect of these multipliers in the Shannon and Tsallis entropies (Cao and Knight 1997; Singh and Luo 2011).
As mentioned, many studies have been done on how to accurately calculate the different types of entropies, the inputs required to calculate the shear stress and the Lagrangian coefficients. Despite these extensive studies, the question arises as to whether entropy models can estimate shear stress in practical projects? Which entropy model is more accurate in estimating shear stress than other entropies? Therefore, the uncertainty analysis of the results of these models can be a bridge between these studies and the selection of the most reliable entropy model for estimating shear stress. For this reason, in this study, only the results of entropy models are analyzed and the focus of this study is not on the initial parameters and differences of the initial assumptions in entropy models. In this study, the uncertainty of four entropy models in estimating shear stress in circular and circular flatbed channels under the same conditions (table 1 in the manuscript file) is evaluated using a new method (HBMES-2). In HBMES, the outputs of entropy models are analyzed, not the inputs of entropy models. (Please see section “3.2”)
- It is hard to find the advancement provided from this study respect to literature studies on the same topics (even those, very recent, from the same group of Authors, i.e. refs 47 and 48). This could a limitation of the present study.
Response: Thank you very much for your comments. Kazemian-Kale-Kale et al. (2018), with the improving uncertainty method used in previous studies, analyzed uncertainty of the Tsallis entropy model in predicting shear stress distribution. They emphasized on the necessity to normalize the shear stress data for the uncertainty analyses. Then, they calibrated the model to select the best sample size (shear stress data considered under different hydraulic conditions) and finally analyzed the uncertainty of the Tsallis entropy model using this sample size (SS). Although their results were well capable of analyzing uncertainty of the Tsallis model, their calibration method was difficult, and they did not discuss the performance of the different transfer functions to normalization in the uncertainty results. Therefore, in another study, Kazemian-Kale-Kale et al. (2020) simplified the calibration method of their previous study, and in addition to the Box-Cox function, Johnson’s function was used to analyze the Shannon entropy uncertainty.
In the present study, the uncertainty of four different entropy models of Shannon, Shannon PL, Tsallis, and Renyi are compared to predict shear stress in the circular channels. The uncertainty prediction is implemented by modifying the uncertainty method presented in previous study, and a novel uncertainty method is introduced. In addition, as a criterion for comparison, the uncertainty of the rgRs model (the common model in prediction of shear stress values) is evaluated as well. At first, the uncertainty method introduced by Kazemian-Kale-Kale et al. (2020) introduced the uncertainty method based on the BMC methodintroduced the uncertainty method based on the BMC method as the Hybrid Bayesian and Monte-Carlo Estimation System (HBMES-1). In this study, the HBMES-1 method to determine whether with a 95% confidence bound (95%CB), entropy models are sufficiently certain to predict shear stress in circular channels or not. The answer to this question is determined by the percentage of measured data within the confidence bound (Nin), but the values of FREE statistics should also be checked for the accuracy of each model relative to the Nin. FREE is a statistical index as equal to the absolute sum of the measured data within the confidence bound (|FP|) and the absolute sum of the measured data outside the confidence bound (|FN|).
It is difficult to compare the certainty of five shear stress prediction models. Therefore, the HBMES-1 method is further improved in this study, and the uncertainty method is presented as HBMES-2. The Minimum CB covers all measured data, OCB (Optimized Confidence Bound) is defined by employing the HBMES-2 method. Then, Based on the OCB, the FREE statistic is optimized and is called FREEopt. Given the value of OCB and FREEopt, the FOCB statistic is introduced that can show the effect of all uncertainty statistics. The drawn OCB represents the width of the confidence bound, which is a quantitative statistic for estimating uncertainty. In addition, FREEopt, which represents the absolute sum of the measured data within the OCB, is a qualitative statistic for estimating uncertainty. FOCB shows the degree of qualitative and quantitative uncertainty of shear stress models. Therefore, it is easy to compare using only one statistic (FOCB) in the HBMES-2 method.
References:
- Kazemian-Kale-Kale, A.; Bonakdari, H.; Gholami, A.; Khozani, Z. S.; Akhtari, A. A.; Gharabaghi, B. Uncertainty analysis of shear stress estimation in circular channels by Tsallis entropy. Physica A 2018, 510, 558-576. https://doi.org/10.1016/j.physa.2018.07.014
- Kazemian-Kale-Kale, A.; Bonakdari, H.; Gholami, A.; & Gharabaghi, B. The uncertainty of the Shannon entropy model for shear stress distribution in circular channels. J. Sediment Res. 2020, 35, 57-68. https://doi.org/10.1016/j.ijsrc.2019.07.001
Round 2
Reviewer 2 Report
The authors improved the original manuscript, but two issues still need to be fixed.
At the end, which method should be preferred in practical cases to estimate channel shear stress ? Please, clarify with one or two sentences the novelty of this study if compared with past studies (Refs. 50 and 51) from the same group of authors.

Author Response
The authors improved the original manuscript, but two issues still need to be fixed:
-At the end, which method should be preferred in practical cases to estimate channel shear stress.
It was done. (Please see conclusion, Lines 663-665)
- Please, clarify with one or two sentences the novelty of this study if compared with past studies (Refs, 50, 51) from the same group of authors.
It was done. (Please see conclusion, Lines 638-642)